# InteracTor: Feature engineering and explainable AI for profiling protein structure-interaction-function relationships

Jose Cleydson F. Silva[1], Layla Schuster[1], Nick Sexson[1], Melissa Erdem[1],
Ryan Hulke[1], Matias Kirst[2,3], Marcio F. R. Resende[4], Raquel Dias[1]*

**1** Department of Microbiology and Cell Science, Institute of Food and Agricultural Sciences, University of Florida, Gainesville, Florida, United States of America, **2** Genetics Institute, University of Florida, Gainesville, Florida, United States of America, **3** School of Forest, Fisheries and Geomatic Sciences, University of Florida, Gainesville, Florida, United States of America, **4** Horticultural Sciences Department, University of Florida, Gainesville, Florida, United States of America

* raquel.dias@ufl.edu

## Abstract

Characterizing protein families' structural and functional diversity is essential for understanding their biological roles. Traditional analyses often focus on primary and secondary structures, which may not fully capture complex protein interactions. Here we introduce InteracTor, a novel toolkit that extracts multimodal features from protein three-dimensional (3D) structures, including interatomic interactions like hydrogen bonds, van der Waals forces, and hydrophobic contacts. By integrating eXplainable Artificial Intelligence (XAI) techniques, we quantified the importance of the extracted features in the classification of protein structural and functional families. InteracTor's interpref features enable mechanistic insights into the determinants of protein structure, function, and dynamics, offering a transparent means to assess their predictive power within machine learning models. Interatomic interaction features extracted by InteracTor demonstrated superior predictive power for protein family classification compared to features based solely on primary or secondary structure, revealing the importance of considering specific tertiary contacts in computational protein analysis. This work provides a robust framework for future studies aiming to enhance the capabilities of models for protein function prediction and drug discovery.

## Author summary

InteracTor is a computational toolkit designed to enhance our understanding of protein structure and function by focusing on three-dimensional (3D) structural interactions. Unlike traditional approaches that primarily rely on sequence or secondary structure data, InteracTor extracts biologically meaningful features such as hydrogen bonds, van der Waals forces, and hydrophobic contacts, which

**Data availability statement:** InteracTor is available as an open-source Python package on GitHub (https://github.com/Dias-Lab/InteracTor), providing a wealth of details in both code and documentation. The repository includes comprehensive implementation details, example datasets, and usage guidelines to ensure transparency and reproducibility. Users can explore, modify, and contribute to enhance its capabilities.

**Funding:** This work was supported by the University of FLorida LIFT AI seed fund to JCFS and RD. The funders had no role in study design, data collection and analysis, decision to publish, or preparation of the manuscript.

**Competing interests:** The authors have declared that no competing interests exist.

are critical for protein stability and dynamics. By integrating these features into machine learning models alongside Explainable AI methods, InteracTor provides interpretable insights into how specific structural interactions influence protein behavior. Our results demonstrate that tertiary structure features significantly improve the accuracy of protein family classification compared to sequence-based methods alone, underscoring the importance of considering 3D interactions in computational protein analyses. The toolkit's modular design makes it adaptable for diverse applications, including drug discovery and protein engineering. In a broader context, InteracTor bridges the gap between computational biology and practical applications in medicine and biotechnology by offering a transparent and robust framework for analyzing proteins at a molecular level. This work represents a step forward in leveraging structural data to advance predictive modeling and biological discovery.

## Introduction

In recent decades, high-throughput sequencing techniques have dramatically expanded protein sequence databases. At the same time, advances in cryo-electron microscopy and deep-learning-based computational structure determination methods, including AlphaFold [1] and RoseTTAFold [2], have transformed protein structure elucidation. Consequently, the surge in available sequence and structural data has catalyzed the development of machine- and deep-learning techniques for predictive modeling. This data has been leveraged to address a variety of challenges, such as identifying non-classical secreted proteins [3,4] predicting binding affinity [5–7], and engineering proteins for novel functions [8,9].

Central to these algorithms is feature engineering and encoding, aimed at converting protein sequences and physiochemical properties into machine-readable formats. Ideally, this process captures the attributes most relevant to the predictive targets of interest. Sequence-based feature representations are among the most widely utilized, including amino acid composition, chemical property-based features, k-mers, and alignment-based embeddings. These descriptors effectively simplify sequence information and reduce the data dimensionality while still highlighting broader functional characteristics, sequence patterns, and evolutionary relationships. However, sequence-based methods can suffer from high dimensionality and data sparsity and are limited in their ability to capture critical properties influencing protein function. The incorporation of 3D structural data into the suite of available encodings allows predictive models to have a deeper layer of biological context that can give insight into functional dynamics. This has spurred the development of more comprehensive feature extraction platforms such as iFeatureOmega [10], Pfeature [11], and ProFeatX [12] which incorporate both sequence and secondary structure descriptors.

Pfeature utilizes amino acid sequences, employing binary encoding for chemical elements and leveraging PaDEL software for fingerprint generation. ProFeatX focuses on

torsional angle bigrams, providing insights into secondary structure. This diverse array of methodologies underscores the complementary nature of these tools, with each excelling in specific bioinformatics applications. However, these tools do not consider the three-dimensional structure of the protein, nor do they consider the interactions between amino acids within the protein's three-dimensional framework. Tools like iFeature harness molecular structures to calculate Half Sphere Exposure (HSE) and Accessible Surface Area (ASA), but still lack support for the extraction of interatomic interactions and other key structural features [10].

While traditional protein sequence descriptors have been fundamental for many predictive modeling types, encoding interaction features remains a significant yet underutilized strategy for capturing the nuances of protein behavior. Recent advancements in sequence- and structure-based embeddings have substantially improved prediction accuracy. However, these embeddings often obscure interpretability and transform even inherently simple models into black boxes by abstracting input features into complex, opaque representations. This makes it difficult to discern how specific biological properties influence model predictions, even when applying advanced explainable AI (XAI) techniques [13]. In contrast, InteracTor's extracted interatomic interaction features are inherently biologically meaningful, allowing for more direct and transparent application of XAI in downstream models. This enables researchers to pinpoint specific interatomic interaction types responsible for observed protein structural and functional properties, offering a clearer path to biological insight.

Here we present InteracTor, a toolkit for the extraction of three types of protein feature encodings: interaction features, physicochemical features, and compositional features. Interaction features include hydrogen bonds, hydrophobic contacts, repulsive interactions, and van der Waals interactions, each encoding unique aspects of molecular dynamics that play an important role in governing protein function. Specifically, hydrogen bonds and hydrophobic contacts are important for stabilizing secondary and tertiary structures [14,15]. Van der Waals interactions influence molecular complementarity, which is crucial for substrate binding, and mediate transient interactions that can facilitate or destabilize protein structures and complexes [16]. Accessible solvent area, hydrophobicity, and surface tension are implicated in protein folding, stability, solubility, and protein-protein interactions. Compositional features include mono-, di-, and tripeptide composition and amino acid side chain chemical property (CPAASC) frequencies [17–19]. These features determine local spatial arrangements (secondary structure) and the overall 3D folded conformation (tertiary structure) of a protein through the formation of alpha helices, beta sheets, loops, and structural motifs.

By leveraging XAI techniques such as Shapley Additive exPlanations (SHAP) [20], we compared the importance of InteracTor's extracted interatomic interaction features to classic primary and secondary structure features across multiple machine learning model architectures. Our feature sets directly map to biologically meaningful concepts, enabling users to readily interpret results and validate the logic of explainable AI models, in contrast to abstract embedding or principal component vectors. This approach allowed us to pinpoint the most impactful features for characterizing protein families and elucidate the relative contribution of tertiary structural information to predictive performance. Structural and sequence-based features were complementary and provided a more comprehensive representation of the protein. Our integrative feature encoding and selection approach underscores the complexity and richness of proteins, ultimately advancing our ability to characterize proteins for various applications in structural biology, biotechnology and medicine.

## Results

InteracTor is an open-source computational toolkit for feature engineering and XAI that operates through four major steps (Fig 1): (1) extraction of atom, residue, and sequence information from protein structure files, (2) calculation of interatomic interactions and physicochemical properties, (3) computation of sequence-based compositional features such as mono-, di-, and tripeptide frequencies, and (4) integration and export of these multimodal features for downstream analysis and interpretation.

### Multimodal protein profiling

InteracTor computes 11 different interatomic interaction features and 8 distinct CPAASC that are key for characterizing structure and function of proteins (Table 1 and S1 Table). In addition to interaction and structural features, our toolkit

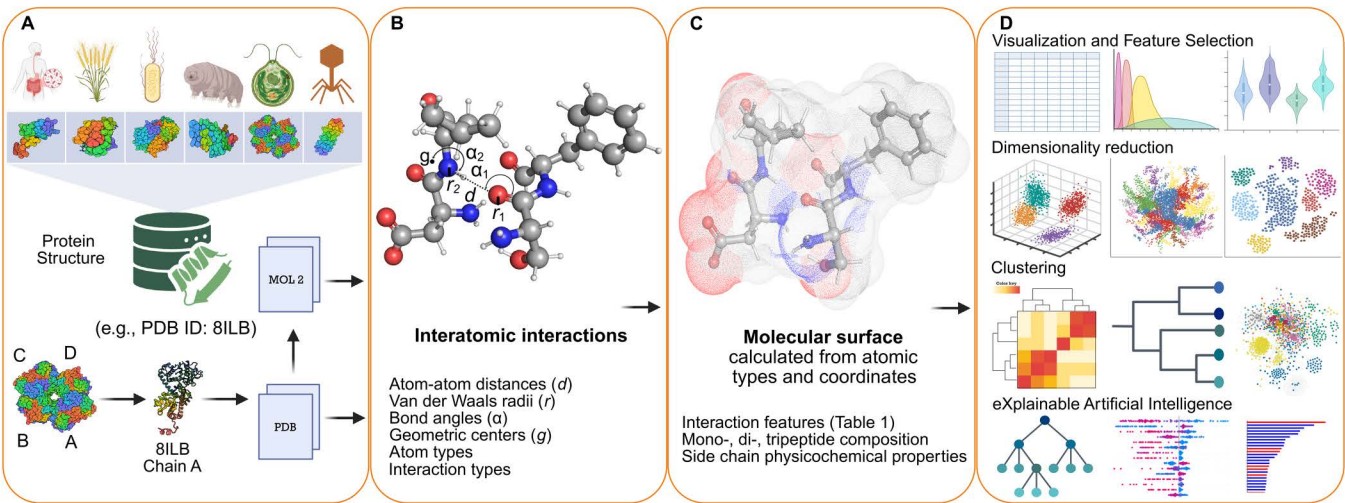

**Fig 1. Pipeline of the interactor algorithm. (A)** Extensive database of PDB files, where each chain is split into an individual PDB file and converted to MOL2 format. **(B)** The following metrics are calculated: atom-atom distances ($d$) and van der Waals radii ($r$); bond angles ($a$); geometric centers ($g$); interaction types and atom types. Molecular representation with carbon atoms in gray, oxygen in red, and nitrogen in blue. **(C)** The following features are calculated: interatomic interaction features (Table 1); side chain physicochemical properties (S1 Table); mono-, di-, and tripeptide composition (S2 Table). **(D)** The extracted features are exported in tabular format, which then are utilized for feature selection, dimensionality reduction, clustering, visualization, and machine learning.

also extracts classic protein sequence compositional features such as k-mer frequencies (S2 Table), resulting in a total of 18,296 multimodal features extracted (Table 2). This provides a comprehensive representation of protein structure, function, and sequence characteristics, enabling in-depth analysis of protein properties across various scales of molecular organization. While we demonstrate InteracTor's utility through protein function family classification, the toolkit is designed for modular adaptation to diverse structural biology tasks—from drug binding analysis to protein engineering—thanks to its interpretable, feature-driven framework.

### InteracTor characterizes the variability among distinct protein families

We conducted principal component analysis (PCA) across all 18,296 features extracted from 20,877 protein structures representing the most abundant protein families and GO terms in the PDB REDO database to evaluate the overall utility of InteracTor in the characterization of variability among protein families and Gene Ontology (GO) terms (S3, S4 Tables). Fig 2 shows the first two of 500 selected PCAs, which captured 1.72% of the variance across protein families (Figs 2A, S1A), and 1.0% of the variance across GO terms (Figs 2B, S1B). Except for Peptidase S1 and the Glycosyl hydrolase 5 (cellulase A), the protein families exhibited well-defined clusters in the two-dimensional space (Fig 2A), whereas GO terms exhibited less separation overall (Fig 2B). This difference may be attributed to the inconsistent accuracy of GO annotations across non-model organisms [30].

Three-dimensional projections provided clearer clustering of the data, allowing a distinct visualization of relationships among the protein families. The first three principal components captured 2.40% of the variance across protein families (Figs 2C, S1A), as well as 1.38% of the variance across GO terms (Figs 2D, S1B). The 3D visualization also improved the separation of Peptidase S1 and Glycosyl hydrolase 5 families, as well as the separation of GO groups, allowing for a more nuanced interpretation of functional relationships among groups.

### Mutual Information scoring identifies key features across protein families

To enhance clustering quality and classification accuracy in downstream analyses, InteracTor performs feature selection using Mutual Information (MI) scoring. All features were ranked by MI score to prioritize those most relevant to the target

**Table 1.** Protein structural features based on interatomic interactions and physicochemical properties.

| Feature Name | Feature Description |
|---|---|
| Total hydrophobic contacts [21] | Aggregation of nonpolar amino acid side chains, which tend to minimize their exposure to the aqueous environment. |
| Total van der Waals (vdW) interactions [5,22] | Weak, non-covalent interactions that arise from transient dipoles induced in atoms or molecules. They are significant when atoms are in proximity, and while individually weak, they collectively contribute substantially to the stability of a protein's structure. |
| Deformation effect [23] | Intrinsic changes and potential movements within the protein that can occur due to its inherent flexibility and dynamic structural nature. |
| Intramolecular hydrogen bonds [24] | Intramolecular hydrogen bonds, formed between electronegative atoms and hydrogen atoms, provide directional and specific interactions that stabilize secondary and tertiary protein structures. We adapted our previously described intermolecular hydrogen bond scoring method to compute intramolecular hydrogen bond scores. |
| Repulsive interactions [22] | Number of atom pairs that are very close to each other, causing their electron clouds to overlap. This overlap leads to a repulsive force due to the Pauli exclusion principle, which prevents electrons from occupying the same space. |
| London dispersion forces [5,25] | The London dispersion forces, a type of van der Waals force, arise from the temporary formation of instantaneous dipoles and contribute to the overall stability of protein structures. While the vdW feature considers interactions within a distance threshold of ≤ 0.7 plus the sum of the van der Waals radii, London dispersion focuses on longer-range interactions, thereby minimizing overlap between these features. |
| Total hydrophobicity [26] | Sum of the hydrophobic contacts within the protein structure, which influence its folding, stability, and interactions. |
| Internal hydrophobicity [26] | Sum of hydrophobic contacts located within the interior of the protein structure. This arrangement is driven by the hydrophobic effect, which is important for protein folding and stability. |
| Total surface tension [27] | Imbalance of attractive intermolecular forces at the surface of a protein, which influences the stability and interactions of protein structures. |
| Internal tension [28] | Resistance encountered by parts of the protein (e.g., secondary structures) as they move relative to each other, essential at all stages of protein folding. |
| Accessible Surface Area (ASA) [29] | Solvent-exposed surface area of a protein, which influences its stability and interactions. |

variables. To optimize feature selection, various MI score cutoffs were systematically tested and compared later in the analysis to identify the subset of features that best balances dimensionality reduction with model performance. Fig 3A shows the distribution of MI scores used for feature selection across protein families. The distribution exhibits a primary mode, or large peak, which corresponds to features with low MI scores (MI < 0.2), reflecting background noise and less informative features. The remaining lower peaks (MI ≥ 0.2) represent a subset of 354 features with informational content effectively distinguished from the background noise peak. Among the top 100 high MI scoring features for the protein family dataset are 9 interatomic interactions, 2 CPAASC features, and 89 sequence composition features (S5 Table). Among the 12 most highly ranked features across protein families (Fig 3B) are intramolecular hydrogen bonds (MI = 0.775), total surface tension (MI = 0.763), London dispersion forces (MI = 0.758), repulsive interactions (MI = 0.722), internal tension (MI = 0.708), Accessible Surface Area (ASA) (MI = 0.694), total hydrophobic contacts (MI = 0.561), dipeptide TG frequency (MI = 0.562), internal hydrophobicity (MI = 0.561), dipeptide VN frequency (MI = 0.556), total hydrophobicity (MI = 0.539), and dipeptide GG frequency (MI = 0.509).

**Table 2. Overview of protein features extracted.**

| Feature type | Number of Features | Feature Description |
|---|---|---|
| Structural features (interatomic interactions and physicochemical properties) | | |
| Interatomic interactions | 11 | Interatomic interactions between residues in a protein, such as hydrogen bonds and hydrophobic contacts. |
| Chemical properties of amino acid side chains (CPAASC) | 8 | The chemical properties of amino acid side chains govern their interactions with other molecules or residues in proteins, encompassing characteristics such as polarity, charge, size, and hydrophobicity. |
| Compositional features (n-peptide descriptors) | | |
| Monopeptide | 26 | Frequency of single residues, or k-mers where k = 1. |
| Dipeptide | 675 | Frequency of two residues linked by a peptide bond, or k-mers where k = 2. |
| Tripeptide | 17,576 | Frequency of three residues linked by peptide bonds, or k-mers where k = 3. |
| Total | 18,296 | |

For GO terms, the MI score analysis revealed values ranging from approximately 0.06 to 0.1, as shown in Fig 3C. While several peaks are well-defined, indicating key features, the MI scores are notably lower compared to those for protein families. Fig 3D presents a violin plot illustrating the distribution of MI scores, with a low density of high MI scores between 0.1 and 0.2. High MI scoring features include total surface tension (MI = 0.202), London dispersion forces (MI = 0.165), total hydrophobicity (MI = 0.146), ASA (MI = 0.142), dipeptide EL frequence (MI = 0.127), total vdW interactions (MI = 0.123), dipeptides EG, NL, KT, GN, and AN, (MI = 0.113, 0.111, 0.109, 0.105, and 0.103, respectively) and repulsive interactions (MI = 0.100). Despite the lower MI scores, these features show a significant association with protein families and GO categories (Wilcoxon test $p \leq 2.2 \times 10^{-16}$).

## Clustering of selected features reveals distinct patterns and relationships among protein families

To uncover inherent structural and functional relationships among protein families based on the selected features, we applied hierarchical clustering analysis to group proteins with similar feature profiles. This clustering approach enables identification of natural groupings and subgroups within the dataset, facilitating downstream interpretation of protein family diversity and functional specialization. The resulting clusters provide a framework for exploring complex inter- and intra-family heterogeneity and complementary subsequent analyses.

Hierarchical clustering grouped the dataset into 19 distinct clusters across eight families (Fig 4A), revealing complex internal organization. The Short-chain dehydrogenases/reductases (SDR) family exhibited the most extensive dispersion, spanning seven clusters (C1, C6, C10, C11, C12, and C13), highlighting its functional diversity. Three distinct clusters (C17, C19, C20) encompassed the Cytochrome P450 family. Similarly, the Peptidase S1 family was distributed across three clusters (C8, C9, C16), suggesting potential functional specialization within this protease group. The Enoyl-CoA hydratase/isomerase family was found in two clusters (C2, C3), indicating possible sub-functionalization. In contrast, several protein families demonstrated a more focused distribution, each confined to a single cluster: Bacterial solute-binding protein 2 (C4), FPP/GGPP synthase (C14), Glycosyl hydrolase 5 (cellulase A) (C15), and Class-I aminoacyl-tRNA synthetase (C18).

Complementary t-SNE clustering analysis identified 9 groups (Fig 4B), not only corroborating these groupings but also unveiling finer details of inter-group relationships. While hierarchical clustering effectively segmented proteins with similar functions, t-SNE provided a more nuanced separation. Occasional cross-family similarities were highlighted with some datapoints from different families appearing in unexpected clusters. Multivariate visualization further supported the role of interatomic interaction features (Fig 4C), which were predominantly observed in the Cytochrome P450 and Glycosyl Hydrolase 5 (cellulase A) families (Wilcoxon $p \leq 4.9 \times 10^{-4}$, S6 Table). In contrast, dipeptide features were more prevalent

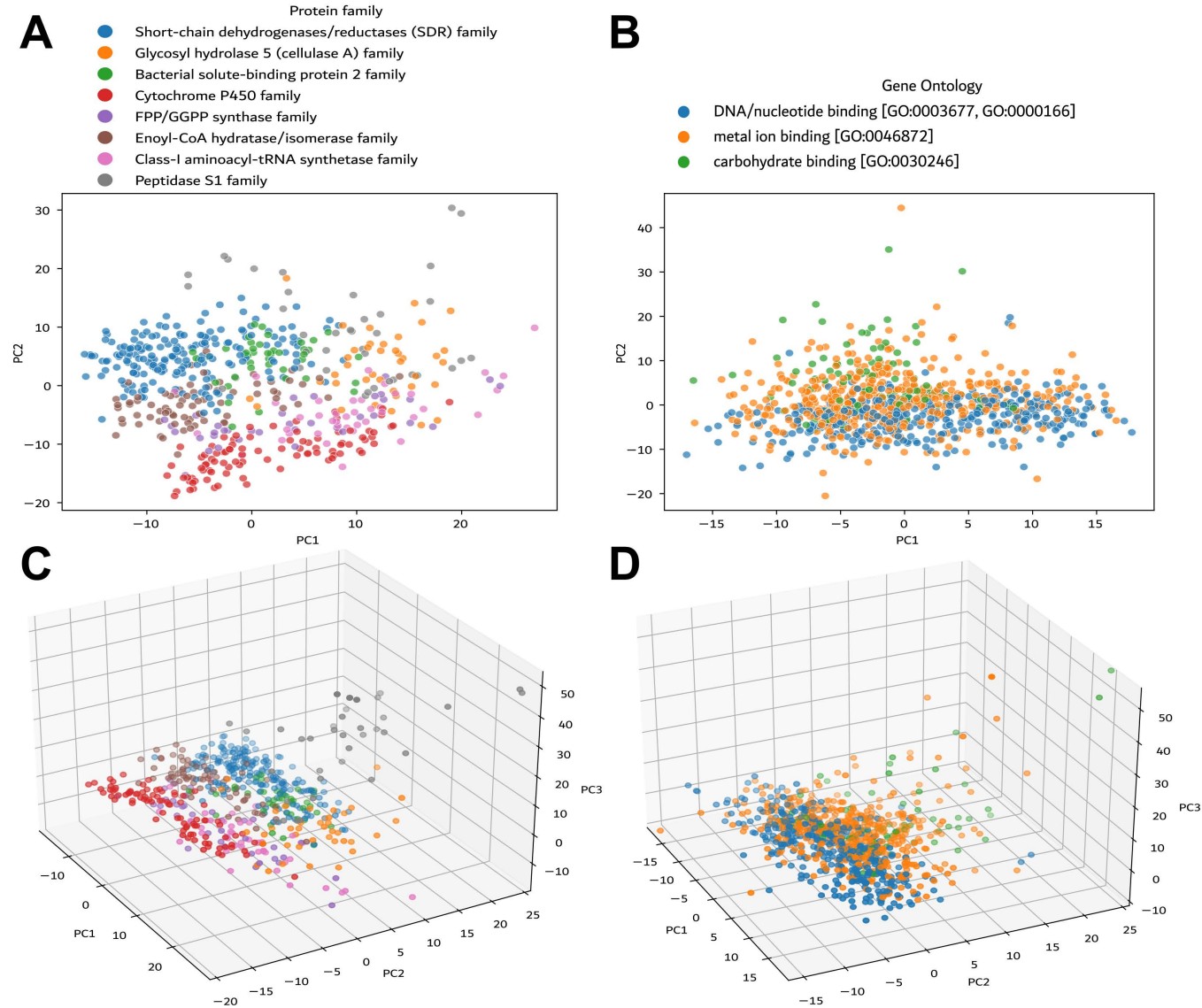

**Fig 2. Principal Cobioremponent Analysis (PCA) of protein families and molecular functions. A)** Clustering of the most prominent protein families. **B)** Clustering of three representative groups based on molecular functions. **C)** Three-dimensional PCA representation of protein families, capturing the variance and complexity within different functional categories. **D)** Three-dimensional PCA visualization of molecular functions.

among other protein families, indicating a varied functional landscape shaped by distinct sequence composition patterns. (Wilcoxon p ≤ 0.031, S6 Table).

### InteracTor's feature selection effectively reduces dimensionality of models without compromising model performance

Protein families exhibited better clustering than Gene Ontology (GO) terms in PCA analysis, so the protein family dataset was selected for downstream model training. Across all feature sets described in Table 2, ensemble models, especially Histogram Gradient Boosting and Random Forest, consistently ranked among the top performers in protein family

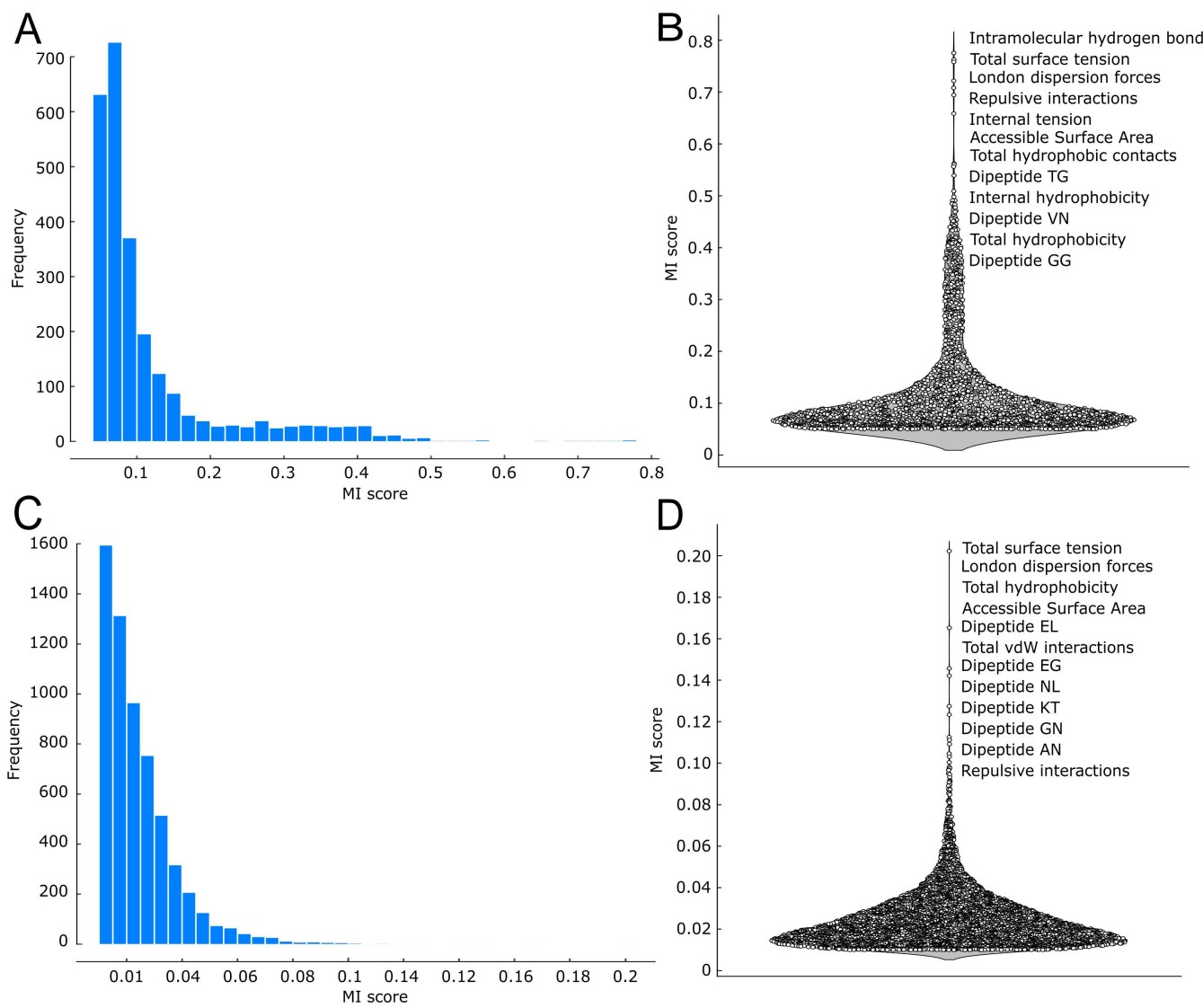

**Fig 3. Mutual Information (MI) scores across protein families and molecular functions. A)** Frequency distribution of MI scores across protein families **B)** Density distribution of MI scores across protein families, focusing on the top 12 ranked features. **C)** Frequency distribution of MI scores for molecular functions. **D)** Density distribution of MI scores for molecular functions, focusing on the top 12 ranked features.

classification, achieving 0.71-0.80 F1-score on the test dataset after 80%-20% train-test split, indicating that the complex relationships captured by InteracTor's features are effectively leveraged by these models' ability to combine multiple decision trees (Fig 5A, S7A–S7D Table). This trend was consistent across different accuracy metrics such as accuracy, precision, recall, and MCC (S7B, S7C Table). CPAASC was the feature type that showed the lowest accuracy across all feature subsets (Tukey's post hoc test p ≤ 0.05, S7A, S8 Tables and Fig 5B). MI score-based feature selection and further F1 score comparison across models showed that reducing InteracTor's feature set to the top 100–500 features achieved comparable performance to using all 18,296 features (Tukey's post hoc test p ≥ 0.82). The smallest feature set selected via MI score (top 100) showed similar performance relative to larger subsets as well (Tukey's post hoc test p ≥ 0.12 versus top 200–500, S8 Table). These results highlight the effectiveness of our feature selection approach for optimizing model performance and interpretability.

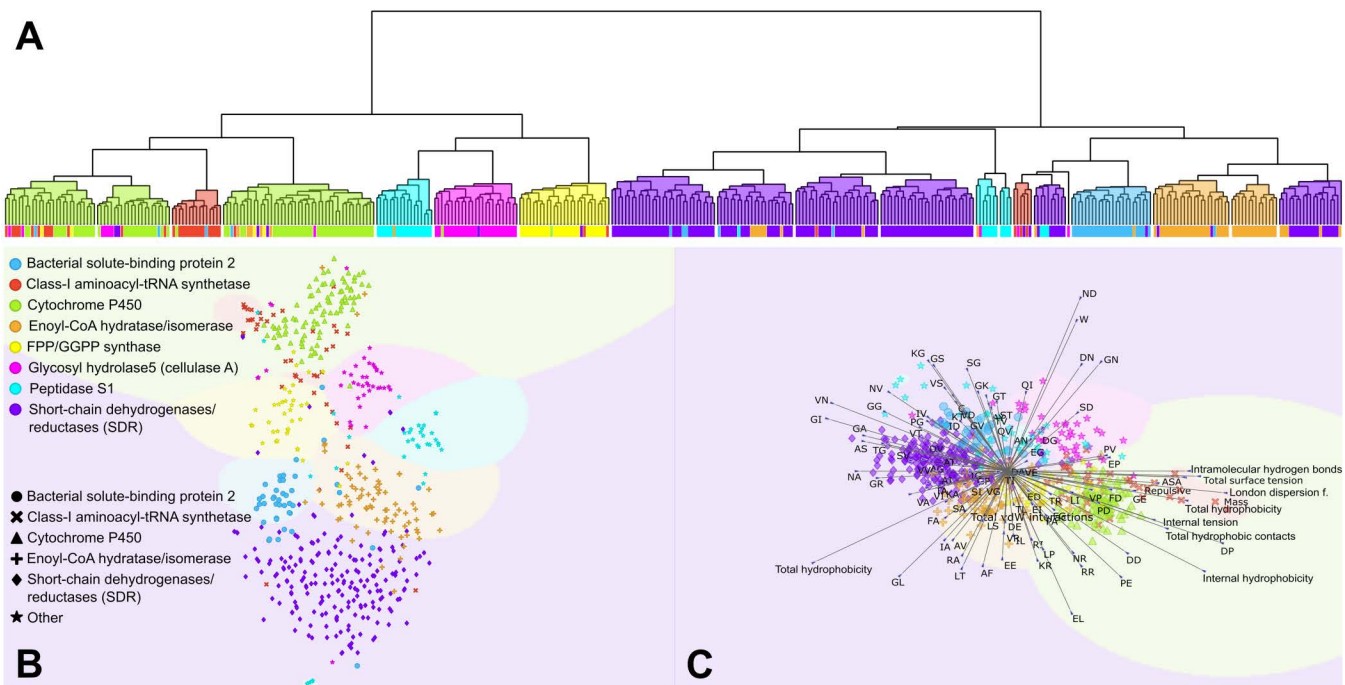

**Fig 4. Clustering and visualization of protein families. A)** Hierarchical clustering of protein families, depicting relationships and groupings based on structural and functional similarities. B) t-SNE plot depicting inter-family relationships within a reduced-dimensional space. **C)** Optimized projection plot highlighting the most significant features for each group, illustrating key attributes that differentiate protein families.

### XAI reveals key features for protein family prediction models

We employed XAI techniques, specifically the SHAP method [20], to analyze the impact of InteracTor's features on machine learning models for protein family prediction. The top 20 features showed SHAP values with heterogenous contributions to the models' predictions across different protein families (Figs 6, S2). The most important features included interatomic interactions and physicochemical properties, such as surface tension and hydrophobicity, as well as specific atomic interactions like repulsive interactions and hydrogen bonds. Additionally, certain peptide composition patterns, such as those involving monopeptides (e.g., P, H, and W) and tri- and di-peptides (e.g., CLG, PP) were also among the most important features. These findings indicate that a combination of multimodal feature types, including structural properties, interatomic interactions, and amino acid composition, critically affects the model's capacity to differentiate protein families.

The features were ranked differently by SHAP values across different protein families, with each protein family type exhibiting a unique association pattern. For instance, London dispersion forces, intramolecular hydrogen bonds, and internal hydrophobicity had a positive impact on predictions for the Cytochrome P450 family (Figs 6A, S2A). However, these same features showed negative impacts for other families, including Bacterial solute-binding protein 2, Class-I aminoacyl-tRNA synthetases, and Short-chain dehydrogenases reductases (Figs 6B–6G, S2B–S2H). FPP/GGPP synthases also exhibited a distinct pattern and ranking of SHAP values relative to other families, with key contributors from dipeptide DD, internal hydrophobicity, London dispersion forces, and several specific amino acid combinations including tripeptide RRG and monopeptide H (Figs 6B, S2B).

To evaluate whether atomic interaction-based features consistently contributed more to model predictions than compositional features across various families, we conducted a two-sample Kolmogorov–Smirnov (KS) test on the total SHAP values for both feature groups. The analysis demonstrated statistically significant differences, with false discovery rate (FDR)

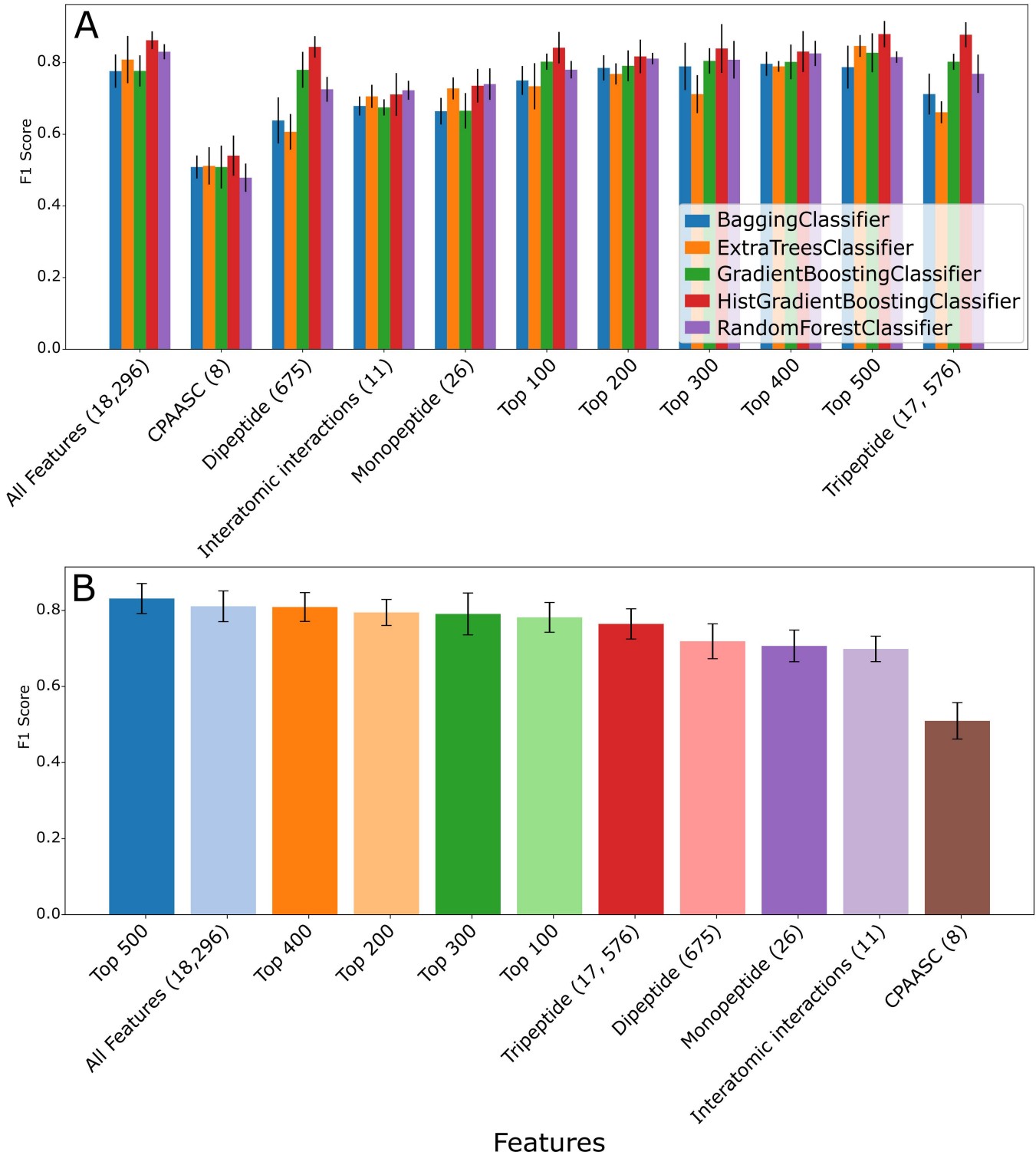

**Fig 5. Machine learning benchmark across different feature sets. A)** F1 scores were computed for several classification models (shown as colored bars) across different feature sets. Feature sets were ranked by their impact on model performance, with higher F1 scores indicating superior predictive accuracy. **B)** Performance of the best model (Histogram Gradient Boosting) across different feature sets, showing mean validation accuracy after 5 runs.

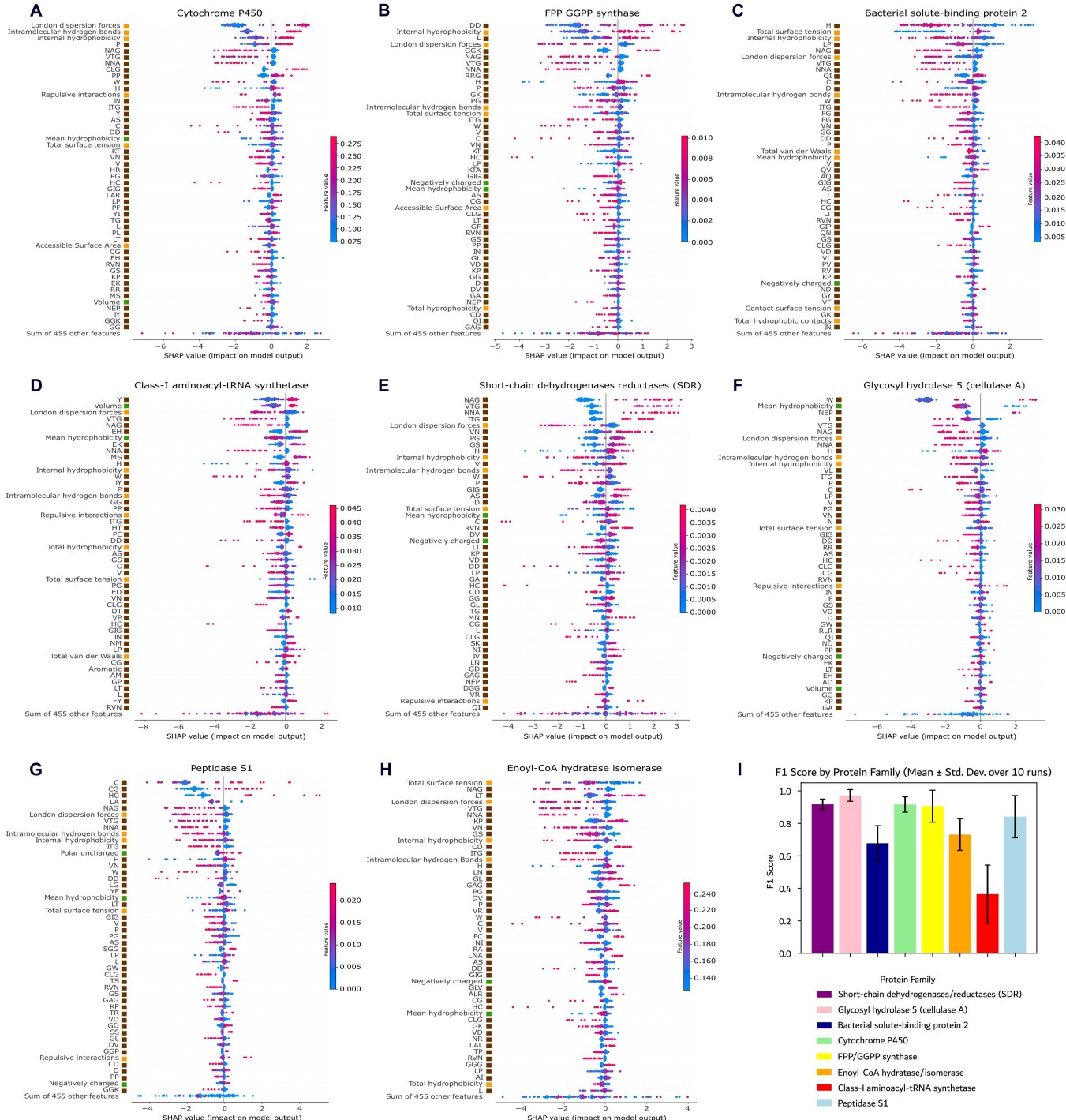

**Fig 6. Feature importance (SHAP) in protein family classification. A-H)** The plot ranks features by their impact on model predictions, illustrating contribution distributions. Each dot represents a protein, with colors indicating feature values: red for high, blue for low. The horizontal spread of dots reflects the density of examples at given feature values. Negative SHAP values (left) indicate a reduced likelihood of classification, while positive SHAP values (right) suggest an increased likelihood. Colored tiles beside each feature indicate feature types: orange for 3D structural features (based on interatomic interactions and structural properties), green for CPAASC, and brown for sequence compositional (n-peptide) features. SHAP values were computed using family-specific explicands and baselines, ensuring consistent cross-group comparisons. **I)** Classification model performance measured by the F1 Score for each protein family. The bars represent the mean F1 Score obtained from 10 independent runs, with error bars corresponding to the standard deviation.

adjusted p-value < 0.001 in all eight protein families and KS statistics spanning 0.45 to 0.82. Glycosyl hydrolase 5 (cellulase A) and Peptidase S1 families presented the greatest separation between interaction and non-interaction features (KS = 0.82 and 0.75, respectively). In contrast, the Cytochrome P450 family showed a lower effect size (KS = 0.45), yet retained strong statistical significance (adjusted $p = 1.6 \times 10^{-19}$). These results underscore that interaction features are broadly enriched for predictive capacity and emphasize their distinct, family-specific importance across protein families (S9 Table).

## Discussion

### A novel algorithm for improved characterization of protein structural properties

Our protein feature extraction algorithm distinguishes itself by focusing on the tertiary structure of proteins, unlike other approaches that rely on primary and secondary structures [10,31,32]. By analyzing three-dimensional relationships, the algorithm captures intramolecular interactions that underpin protein structure and function. Extracting features directly from the tertiary structure allows the algorithm to uncover additional patterns in the chemical properties of amino acids and their interactions, offering a comprehensive understanding of the structural and functional dynamics of proteins [33]. This enhanced perspective provides new insights into the complex mechanisms governing protein behavior, facilitating advancements in protein engineering and drug design [34]. The graphical representations (Figs 3–6) provide a comprehensive overview of protein characteristics as well, assisting in the visualization and interpretation of complex structural and physicochemical properties across protein families.

The feature selection process, guided by MI scores, was crucial in identifying the most effective features for distinguishing protein families (Fig 3). Prominent peaks in MI scores indicated that features such as hydrogen bonds, total surface tension, and contact hydrophobicity are particularly influential in differentiating protein families. This prioritization of key features enhances our ability to effectively distinguish between various protein structural and functional groups. The identification of 9 out of 11 interatomic interactions features among the top 12 features with the highest MI scores underscores their importance in differentiating protein families. This finding highlights the importance of specific molecular interactions, such as hydrogen bonds and hydrophobic contacts, in defining protein structure and function, providing valuable insights into the fundamental principles governing protein family diversity.

Notably, several dipeptide-based descriptors, including TG, VN, and GG, were identified among the top-ranked features in our mutual information analysis. Given the much larger number of possible tripeptides compared to dipeptides, we would expect more tripeptides than dipeptides to appear among the top features, reflecting a feature representation bias. However, MI scoring inherently adjusts for feature frequency and only highlights those patterns that occur often enough to be reliably informative. This helps account for feature representation bias, so the overrepresentation of certain n-peptides among top predictors reflects their actual predictive value with the available sample size, rather than simply their greater number. Biologically, tripeptides capture more specific local sequence contexts and functional motifs that are crucial for molecular recognition, enzymatic activity, and binding interactions [35,36]. Therefore, beyond statistical considerations, the prominence of some tripeptides among top features likely reflects their genuine molecular and functional importance.

Fig 4 shows clear separations of specific clusters, highlighting the effectiveness of the approach in identifying grouping patterns among protein families. The groupings reflect complex interatomic relationships between different features and demonstrate the formation of distinct clusters within dipeptide feature groups. The Peptidase S1 family, Glycosyl hydrolase 5 (cellulase A) family and Short-chain dehydrogenases/reductases (SDR) family demonstrate clustering of dipeptide feature groups. This result confirms the method's ability to highlight relevant functional patterns and distinguish between different protein family categories based on the observed interatomic interactions.

### Clustering reveals complex structure and functional diversity in protein families

Hierarchical clustering effectively delineated differences among protein families such as short-chain dehydrogenases/reductases (SDR), Cytochrome P450, Enoyl-CoA hydratase/isomerase, Bacterial solute-binding protein 2, Class-I

aminoacyl-tRNA synthetase, Glycosyl hydrolase 5 (cellulase A), Peptidase S1, and FPP/GGPP synthase. Complementary analysis with t-SNE allowed for the detection of subtle similarities and differences among the protein families and offered deeper insights into their functional connections (Fig 4, S6 Table). Multivariate visualization allowed for the identification of unique aspects of protein families, including specific features contributing to the clustering process, yielding valuable insights into the functional diversity within and between protein families. This integrative approach led to a deeper understanding of the complex relationships and functional diversity inherent across protein families, revealing details on the interplay of intermolecular interactions, physicochemical properties, and composition of proteins. This holistic methodology not only enhances our comprehension of protein family dynamics but also establishes a robust framework for future research. The extracted features have the potential to directly impact the performance of machine learning and deep learning models by offering a new feature engineering and feature selection tool. By providing new insights on protein structural and functional properties, our approach supports advancements in structural biology, synthetic biology and in the characterization proteins of unknown function (PUFs) [37–40]. Additionally, these features can contribute to emerging fields such as quantum biology [41–45], facilitating a deeper understanding of biological roles and interactions across diverse protein families.

## XAI reveals key features for classification of protein function families

Our XAI results show that a combination of physicochemical properties, atomic interactions, and peptide motifs enhances protein family classification accuracy. This approach underscores the importance of integrating protein structural, physicochemical and sequence features in computational modeling, assembling groundwork for advanced protein analysis methods. By integrating these features, we open a new pathway for understanding protein structure, activity, and function. Feature selection and XAI analysis revealed that atomic-level interactions, specifically hydrogen bonds and London dispersion forces, in conjunction with distinct peptide composition patterns (e.g., proline, cysteine-leucine-glycine, and diproline sequences) were pivotal for identifying protein families. Moreover, the application of XAI methodologies elucidated the differential impact of these features across various protein families (Fig 6).

Our findings align with previous studies that emphasize the significance of atomic interactions and physicochemical properties in protein classification [5,10–12,17–19]. Other sequence based features have been widely reviewed [46,47]. Our proposed set of interatomic interaction features were often among the top ranked features contributing to the accurate classification of protein families, complementing other feature types (S5–S9 Tables). By leveraging XAI, we successfully quantified and interpreted the specific impact of each feature- an aspect that conventional approaches could not comprehensively address- offering deeper insights into the biological mechanisms underlying the model's performance.

The capacity to identify key features for protein classification has implications for drug design and protein engineering because understanding molecular interactions is critical. InteracTor enables interpretable modeling tasks such as protein feature engineering, feature selection, and XAI, supporting applications like drug discovery (e.g., binding site analysis), protein function prediction, and structure-guided protein design. InteracTor applies XAI principles—such as feature importance attribution and structural interpretability—to map interatomic interactions to functional outcomes. This bridges the gap between black-box protein models and actionable insights, enabling targeted protein engineering (e.g., optimizing binding affinity or stability by prioritizing CPAASC features) and error diagnosis (e.g., flagging repulsive vdW clashes in misfolded designs).

The consistent enrichment of SHAP values associated with atomic interaction features across all protein families suggests these interactions encapsulate fundamental aspects of protein architecture (e.g., folding stability, conformational constraints, and domain compaction). Unlike conventional sequence-derived descriptors, which typically reflect evolutionary conservation or motif recurrence, interaction features provide a direct readout of the structural energetics underlying protein behavior. By leveraging a model-agnostic statistical framework to quantify the influence of these features, we offer

rigorous, reproducible evidence for their importance. This approach paves the way for generalizing interaction-aware modeling to a wider range of protein classification tasks and underscores the foundational importance of structural biophysics in computational protein science.

## Mechanistic insights on protein structure, activity, and function

The features extracted by InteracTor provide critical insights into the structural and functional diversity of analyzed protein families. Hydrophobicity-related features consistently rank among the top predictors across most families; however, internal hydrophobicity is particularly important for classifying the Cytochrome P450 and FPP GGPP synthases, while mean hydrophobicity plays a more prominent role in other protein families (Figs 6, S2). The variation in feature importance can be attributed to the structural and functional diversity of the protein families. For example, the importance of London dispersion forces [48] and internal hydrophobicity [49] in Cytochrome P450 classification [50] emphasizes the crucial role these interactions play in maintaining its tertiary structure (Fig 6A). The tripeptides NAG and NNA showing as the top 1 and top 3 predictors in Short-chain dehydrogenases reductases (SDR) are consistent with the conserved NNAG motif, known for stabilizing the β-strands within the central β-sheet of the characteristic Rossmann fold that is critical for cofactor binding and enzymatic function in this family (Fig 6E) [51,52]. The tripeptides VTG and ITG showing as the top 2 and top 4 redictors are also consistent with the glycine-rich cofactor-binding motif (TGxGxxxG sequence) in the SDR family [51]. These peptide features likely capture key conserved elements of the NAD(P)(H) binding site, reflecting their important structural and functional role in SDR enzymatic activity [53]. These family-specific variations highlight how different feature types contribute distinctively and synergistically to protein classification.

Intramolecular hydrogen bonds were also among the top predictors for classifying SDR proteins (Fig 6E). Hydrogen bonds are known to be essential in the stabilization of the Rossmann fold and the NAD(P)(H)-binding region of SDRs, which is consistent with our findings [54,55]. ASA and total hydrophobic contacts were key features in the classification of Cytochrome P450 family (Fig 6A). The importance of hydrophobic contacts in Cytochrome P450 function has been highlighted by studies showing that hydrophobic residues are pivotal in complex formation with their redox partners [56]. Cytochrome P450s also have deeply buried active sites that are connected to the solvent by a network of channels exiting at the distal surface of the protein, which could be reflected by the high contribution of ASA in the classification of this protein family [57].

The Bacterial solute-binding protein 2 family displayed a balance of total surface tension and internal hydrophobicity, showing equally intense but opposite effects (Fig 6C), which could reflect in the dynamic conformational states involved in substrate transport [58,59]. While this finding is intriguing, further studies involving other dynamic protein families are necessary to confirm whether this behavior is a generalizable feature. Hydrophobicity-related features were key predictors for the Peptidase S1 family (Fig 6G). Chymotrypsin-like enzymes within the S1 family have a hydrophobic S1 pocket, which allows them to cleave peptide bonds following medium to large hydrophobic amino acids such as tyrosine, phenylalanine, and tryptophan, which is consistent with our findings [60]. The Peptidase S1 family also contains a catalytic site that is typically preceded by a block of hydrophobic residues as well, which is also consistent with our results [61]. These observations highlight InteracTor's utility in characterizing protein families based on their global structural features, offering a foundation for future studies exploring interaction-specific mechanisms within these families.

## Methods

### Algorithm overview

InteracTor algorithm consists of a sequence of steps for analyzing protein structures and calculating various features from proteins' interatomic interactions, physicochemical properties of residues, and peptide composition (Fig 1). The following is a description of each step:

*Step 1.* Extract atom, residue, and sequence information from PDB file (Fig 1A): this process involves parsing the Protein Data Bank (PDB) file to obtain the atomic types, 3D coordinates, and the amino acid sequence of the protein.

*Step 2.* Extract additional atomic information from MOL2 file (Fig 1A), including covalent bond mapping and types (e.g., single, double, triple bonds, aromatic rings, etc.) and $sp^2$ hybridization.

*Step 3.* Calculate atom-atom distances (Fig 1B): the algorithm computes van der Waals (vdW) radii and distances between pairs of atoms not covalently bound within the protein structure, which is used to identify potential interatomic interactions within the protein structure. A distance threshold, as described by Dias et al., [5] is applied to determine whether atoms are interacting.

*Step 4.* Calculate geometric centers (Fig 1B): additional properties are extracted from atoms selected as potentially participating in non-covalent interactions (step 3) and covalently bound to multiple atoms (step 2). This includes the calculation of geometric centers between the selected atoms and their respective covalently bound atoms. These centers are utilized to calculate angles between hydrogen bond donor and acceptor atoms (step 5).

*Step 5.* Calculate angles (Fig 1B): the algorithm then computes the angles between atoms or groups of atoms in the protein structure, which is used to evaluate hydrogen bonding geometry.

*Step 6.* Compute interatomic interaction features and protein physicochemical properties described in Table 1 and Fig 1C.

*Step 7.* Extract compositional features (S2 Table and Fig 1C): InteracTor calculates n-peptide frequencies such as mono-peptides, dipeptides, and tripeptides in the protein sequence.

*Step 8.* Extract CPAASC frequencies (S1 Table and Fig 1C): the algorithm computes the frequencies of several physico-chemical properties associated with the side chains of the amino acids in the protein sequence.

*Step 9.* Write results and postprocessing (Fig 1D): the final step is to write the features computed in steps 6, 7 and 8 to an output file for further analysis or use in other applications. The toolkit also includes example scripts for downstream analyses such as feature selection via MI scoring, dimensionality reduction via PCA, t-SNE, UMAP, hierarchical clustering and visualization using heatmaps, and protein function classification using machine learning and XAI.

Our toolkit extracts and encodes protein structural features based on physico-chemical properties, amino acids composition, and interatomic interactions. By building upon our previous methods for predicting protein-ligand binding affinity [5], we modified the algorithm to analyze residue-residue interactions within protein structures (Table 1). We also included the calculation of CPAASC frequencies [17], (S1 Table), monopeptide [18], dipeptide, and tripeptide composition [19] (S2 Table). In addition to the 20 classic amino acids, our algorithm supports rare residues found in distinct biological systems, including selenocysteine, pyrrolysine, and N-methylvaline. These amino acids, although rare, are key in specific biological processes [62,63].

## Structural datasets and preprocessing

*Dataset for clustering analyses.* We used 20,877 protein structures from the PDB-REDO database [64], having eliminated redundant (50% sequence similarity) and small proteins (<50 residues) using Biopython and in-house scripts (see Data availability statement). PDB-REDO was used to enhance structural data quality, improving machine learning model reliability by minimizing errors and inconsistencies (e.g., low resolution, missing atoms, etc.). This approach aligns with research showing that prioritizing data quality over quantity leads to better prediction performance and model robustness [64]. Including distant protein relatives with lower identity cutoffs in the study can provide valuable insights into functional conservation and evolutionary relationships as proteins with low sequence identity can still share similar tertiary structures and functions [64]. PDB files were converted to MOL2 format using Open Babel [65]. We then applied the InteracTor algorithm to extract features from both PDB and MOL2 files.

*Datasets for machine learning and XAI.* We further filtered out proteins with more than 50% sequence identity to minimize data leakage in our machine learning and XAI experiments. We also filtered out structures with low resolution (>2.5 Å) and less than 100 residues in order to reduce noise and potential outliers.

*Datasets selected for demonstration.* We utilized UniProt's application programming interface (API) [66] to extract Gene Ontology [67] (GO) terms and protein family names based on the PDB accession number [68] to facilitate analysis and demonstration of use cases. We performed power analysis to determine the minimum sample size needed per protein family and GO term for further demonstration of use cases (Tables 1 and 2). For the demonstration of use cases, we selected protein family categories with at least 30 representatives in our dataset to ensure sufficient statistical power for downstream analyses. Similarly, we applied a minimum threshold of 90 annotations for Gene Ontology (GO) terms but further restricted selection to terms directly associated with the binding mechanisms of proteins (e.g., ligand type) to further assess the potential of InteracTor for profiling of protein functions directly associated with ligand or substrate binding. By selecting protein families and GO terms with enough sample sizes, we ensure that our use cases effectively demonstrate our approach's capabilities. Mapping and power analysis scripts are available in GitHub as well (see Data availability statement).

## Feature selection and clustering analysis

We applied MI scoring to quantify the relevance of each feature in distinguishing between different protein families and GO categories [69]. We ranked the features by their respective MI scores and selected the top 100 most informative features for further analyses.

We applied PCA to reduce the dimensionality of the data generated by our toolkit and evaluated how the primary components capture the variability among protein families and GO terms [70]. In addition to PCA, we also applied t-SNE [71,72] and Freeviz [73] to reduce dimensionality while preserving nonlinear relationships and local structures in data. We used violin plots to visualize the distribution and density of the extracted features across different protein families and GO terms, providing insights into the distribution and central tendencies of the data.

We performed hierarchical clustering using Pearson correlation as the distance metric and the complete linkage method for cluster formation [72]. This approach allowed us to identify and visualize the hierarchical relationships between protein families based on similarities among the features extracted by our toolkit. The dimensionality reduction and clustering results were visualized with FreeViz.

## Machine learning benchmark

We generated 11 feature sets to assess their contribution to improving machine learning performance in predicting structural properties. Five datasets comprised individual feature types: interatomic interactions, CPAASC, monopeptides, dipeptides, and tripeptides (Table 2). Another five datasets were created using the top 100, 200, 300, 400, and 500 features ranked by MI scores, incorporating a mixture of the five feature types (S5 Table). The eleventh dataset integrated all features extracted by InteracTor (Table 2). This approach allowed us to evaluate the relative importance of different feature combinations in enhancing predictive accuracy.

We tested Machine Learning (ML) models to assess the performance of InteracTor's features in the prediction of protein families. In this experiment, we used 43 ML algorithms implemented in the Python library Scikit-learn (S7D Table) [74]. To evaluate model performance, we randomly shuffled the data, performed 80%-20% train-test split and measured multiclass classification accuracy, precision, F1 score, Mathew's correlation coefficient (MCC), and precision for each combination of feature set and algorithm [72]. Feature selection was conducted exclusively on the 80% training set to identify the most informative features.

## eXplainable AI (XAI) methods

We calculated SHAP [75] values to quantify the individual feature contributions in predicting protein families using best-performing model (HistGradientBoostingClassifier). SHAP values were extracted from the test set using the trained model. We then generated SHAP summary plots using the summary plot function from the SHAP library [20]. To create SHAP bar plots, we calculated the total absolute SHAP values for each feature and determined the effect direction by computing the Pearson correlation between SHAP values and the model's predicted probabilities

## Statistical analysis

Statistical Power tests [76] were performed using the TTestIndPower from the statsmodels package in Python to compute the Cohen's d effect sizes for comparisons between groups (effect size = 0.8, alpha error = 0.05, power = 0.8). The Wilcoxon [77] test was used to compare the means and medians between feature sets and models. We ran Tukey's test [78] pairwise comparisons among F1 scores to identify feature sets that significantly contribute to model performance. We applied p ≤ 0.05 as statistical significance threshold across all statistical tests.

## Features enrichment analysis

To statistically assess the relative importance of 3D interatomic-interaction-based features compared to other compositional descriptors, we performed a Kolmogorov–Smirnov (KS) test on SHAP values across all protein families. For each protein family, we used the SHAP feature importance values generated by the model and split the features into two groups: (1) Interatomic Interaction features; and (2) Other features—comprising all remaining compositional and structural descriptors used in the model.

We computed the sum of SHAP values per protein for each feature group (interaction vs. others), thus obtaining two distributions of summed SHAP values per family. These distributions were then compared using a two-sided two-sample Kolmogorov–Smirnov test, as implemented in scipy.stats.ks_2samp. This non-parametric test evaluates whether the cumulative distributions of the two feature sets differ significantly. A large KS statistic (D) and a small p-value indicate that the interaction features, as a group, contribute differently—and often more strongly—to the model's predictive behavior compared to the other features. To correct for multiple testing across the eight protein families, we applied the Benjamini–Hochberg procedure to control the false discovery rate (FDR), although Bonferroni correction was also tested and yielded similar significance patterns.

## Supporting information

**S1 Fig. Principal component analysis of protein families and GO terms.** A) Scree plot illustrating the variance captured by each of the 500 principal components across different protein families. B) Scree plot depicting the variance explained by each of the 500 principal components across Gene Ontology (GO) terms.
(TIFF)

**S2 Fig. Global feature importance in Protein Family Classification.** Mean absolute SHAP values were computed for each protein family, representing the overall impact of features on each protein family. The direction of the impact was computed based on the correlation between the SHAP values and the likelihood of the predicted class: red for positive correlation, blue for negative correlation. Subplots (A-H) correspond to distinct protein families. Colored tiles beside each feature indicate feature types: orange for 3D structural features (based on interatomic interactions and structural properties), green for CPAASC, and brown for sequence compositional (n-peptide) features. I) Learning curve for the best model (Histogram Gradient Boosting), showing validation F1 score as a function of training sample size.
(TIFF)

**S1 Table. Protein physicochemical features based on chemical properties of amino acid side chains (CPAASC).**
(DOCX)

**S2 Table. Protein sequence composition features.**
(DOCX)

**S3 Table. Distribution of selected protein families.**
(DOCX)

**S4 Table. Distribution of selected GO terms.**
(DOCX)

**S5 Table. MI Scores for protein families and GO terms.**
(XLSX)

**S6 Table. Wilcoxon p-values for pairwise comparisons between protein families.**
(XLSX)

**S7 Table. A) Ranking of models trained on different feature sets across all tested algorithms, sorted by their F1-score.** B) Ranking of models grouped by algorithm type, based on their average F1-score across different feature sets, providing insights into relative algorithm effectiveness. C) For each feature set, the top five models (algorithm-feature set combinations) with the highest F1-score are highlighted to identify the best-performing pairs. D) A detailed overview of algorithm characteristics and the performance of models based on these algorithms, including descriptions of their principles, strategies, and associated metrics.
(XLSX)

**S8 Table. Tukey's post-hoc test comparing validation F1-scores of models trained on different feature sets, highlighting statistically significant differences in model performance.**
(XLSX)

**S9 Table. Feature enrichment test by two-sided two-sample Kolmogorov–Smirnov test.**
(XLSX)

## Acknowledgments

The authors gratefully acknowledge UF Research Computing for providing computational resources and support that have contributed to the research reported in this publication (http://www.rc.ufl.edu).

## Author contributions

**Conceptualization:** Jose Cleydson F. Silva, Layla Schuster, Raquel Dias.

**Data curation:** Layla Schuster, Nick Sexson.

**Formal analysis:** Jose Cleydson F. Silva, Raquel Dias.

**Funding acquisition:** Raquel Dias.

**Investigation:** Jose Cleydson F. Silva, Layla Schuster, Raquel Dias.

**Methodology:** Jose Cleydson F. Silva, Layla Schuster, Nick Sexson, Raquel Dias.

**Project administration:** Raquel Dias.

**Resources:** Raquel Dias.

**Software:** Jose Cleydson F. Silva, Layla Schuster, Nick Sexson, Raquel Dias.

**Supervision:** Raquel Dias.

**Validation:** Jose Cleydson F. Silva, Melissa Erdem, Ryan Hulke, Raquel Dias.

**Visualization:** Jose Cleydson F. Silva, Layla Schuster, Raquel Dias.

**Writing – original draft:** Jose Cleydson F. Silva, Layla Schuster, Melissa Erdem, Ryan Hulke, Matias Kirst, Marcio F. R. Resende, Raquel Dias.

**Writing – review & editing:** Jose Cleydson F. Silva, Layla Schuster, Melissa Erdem, Ryan Hulke, Matias Kirst, Marcio F. R. Resende, Raquel Dias.

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
