## [Decision Letter · Decision Letter 0]

13 Jul 2025

PCOMPBIOL-D-25-00691

InteracTor: Feature Engineering and Explainable AI for Profiling Protein Structure-Interaction-Function Relationships

PLOS Computational Biology

Dear Dr. Dias,

Thank you for submitting your manuscript to PLOS Computational Biology. After careful consideration, we feel that it has merit but does not fully meet PLOS Computational Biology's publication criteria as it currently stands. Therefore, we invite you to submit a revised version of the manuscript that addresses the points raised during the review process.

Please submit your revised manuscript within 30 days Sep 12 2025 11:59PM. If you will need more time than this to complete your revisions, please reply to this message or contact the journal office at ploscompbiol@plos.org. Please include the following items when submitting your revised manuscript:

We look forward to receiving your revised manuscript.

Kind regards,

Fei Guo

Academic Editor

PLOS Computational Biology

Shihua Zhang

Section Editor

PLOS Computational Biology

**Additional Editor Comments :**

Please revise paper based on reviewers' comments.

**Journal Requirements:**

At this stage, the following Authors/Authors require contributions: Jose Cleydson F. Silva, Layla Schuster, Nick Sexson, Melissa Erdem, Ryan Hulk, Matias Kirst, Marcio F. R. Resende, and Raquel Dias. Please ensure that the full contributions of each author are acknowledged in the "Add/Edit/Remove Authors" section of our submission form.

3) Some material included in your submission may be copyrighted. According to PLOSu2019s copyright policy, authors who use figures or other material (e.g., graphics, clipart, maps) from another author or copyright holder must demonstrate or obtain permission to publish this material under the Creative Commons Attribution 4.0 International (CC BY 4.0) License used by PLOS journals. Please closely review the details of PLOSu2019s copyright requirements here: PLOS Licenses and Copyright. If you need to request permissions from a copyright holder, you may use PLOS's Copyright Content Permission form.

Potential Copyright Issues:

i) Figures 1A, and 1D. Please confirm whether you drew the images / clip-art within the figure panels by hand. If you did not draw the images, please provide (a) a link to the source of the images or icons and their license / terms of use; or (b) written permission from the copyright holder to publish the images or icons under our CC BY 4.0 license. Alternatively, you may replace the images with open source alternatives. See these open source resources you may use to replace images / clip-art:

4) Please amend your detailed Financial Disclosure statement. This is published with the article. It must therefore be completed in full sentences and contain the exact wording you wish to be published.

3) If any authors received a salary from any of your funders, please state which authors and which funders.

**Reviewers' comments:**

Reviewer's Responses to Questions

Reviewer #1: The work presented here describes a predictive ML method for the structural classification of protein families and their function.

Several aspects, namely the use of 3D structures and interpretable models, are two of the strengths of this work.

Overall, the manuscript covers all the methodological details of the work with sufficient clarity, and approaches appear to be rigorous and robust.

I do have some minor reserves mostly related to the results analysis and the implications of the features deemed responsible of the model performance, and minor wording clarifications throughout the manuscript.

In particular, the correlation between features role and protein classes feels rather anecdotal and week. This part could be strengthen very easily by finding more examples and consistent outcomes across protein families (i.e.: all/the majority of transporters show features X and Y ).

Below a few more in-detail comments.

- vdW vs London forces: interesting take, but how did they calculate them with sufficient accuracy to differentiate them and prevent double-counting?

- Out of the features extracted, how the Authors addressed the dominance of features describing the frequency of a tripeptide? While not being at risk of providing unwanted memorization of the dataset, it still represents a dominant set of features that capture the sequence more than the structural context of the amino acids, arguably one of the strengths of the work.

In that regard, I assume that in the Mutual Information analysis, the descriptors of TG, VN and GG descriptors refer to the frequency of dipeptides of such amino acids. If that's correct, it's interesting that dipeptide descriptors were found more important than tri-peptides?

> Additionally, certain peptide composition patterns, such as those involving amino acids P, CLG, PP, H, and W, were also among the most important features.

This is somehow a recurrent pattern in the manuscript, where amino acid one-letter references are mixed with other acronyms in a not so clear manner. Also, according to the nomenclature introduced earlier, those should be referred to as n-peptide descriptors, not "amino acids".

> For instance, the importance of London dispersion forces(45) and internal hydrophobicity(46) in Cytochrome P450 classification emphasizes the crucial role these interactions play in maintaining its tertiary structure

This statement and its inverse in the following paragraph are a bit controversial. Hydrophobicity is considered among the most important forces driving protein folding, which would be expected to hold true for virtually every protein in the dataset, and not in specific classes.

> Bacterial solute-binding protein 2 family displayed a balance of repulsive interactions and internal tension, likely linked to their dynamic conformational states required for substrate transport(54)

This is an interesting finding, but unless confirmed across multiple dynamic proteins, it really falls a bit short of supporting the post-hoc analysis discussed.

Also, as a suggestion for future work, it would be very interesting to see the performance of a model extended to include secondary structure descriptors.

Reviewer #2: This paper by Silva et al explores the classification of protein family and protein function using machine learning methods. The specific contribution of this work is the use of features based on 3D protein structures. These features are based on, for example, patterns of hydrogen bonding, hydrophobic contacts, and van der Waals interactions.

Overall, this work is scientifically interesting and rigorously conducted. I have only three comments for the authors' and editors' consideration:

First, this work is based on the 20,877 protein structures in the PDB REDO data set. The manuscript provides good justification for using this subset of the entire PDB based on data quality and data leakage. But the thresholds used only leave 8 protein families and 3 GO terms, which limit the generalizability of the conclusions. Given that the GO analysis uses a higher threshold for the sample size (90 vs 30), the authors should consider relaxing that threshold to increase the number of GO terms in their analysis.

Second, Figure 5 shows the F1-score on subsets of features as they compare to the full set of features. It appears this evaluation was done based on a 80-20 split, where 80% of the samples are used to train the classifier and 20% are used to calculate the F1-score. I infer that the feature selection was also done on the 80% training set. I believe that the more rigorous approach would be a three-way split of samples, with one set used strictly for the feature selection. This design would better support the generalized assertion that the feature selection method produces comparable classification results.

Third, in Figure 6, consider highlighting the 3D interaction features (by bolding or coloring) that are the focus of this study. Especially since the feature lists are heavily dominated by the compositional features, visually highlighting those interaction features will reinforce their importance. To assign a statistical significance, consider also using a Kolmogorov-Smirnoff statistic to show that these interaction features have larger SHAP values (similar to how the GSEA tool performs gene set enrichment).

**Have the authors made all data and (if applicable) computational code underlying the findings in their manuscript fully available?**

Reviewer #1: Yes

Reviewer #2: Yes

PLOS authors have the option to publish the peer review history of their article (what does this mean? ). If published, this will include your full peer review and any attached files.

**Do you want your identity to be public for this peer review?** For information about this choice, including consent withdrawal, please see our Privacy Policy .

Reviewer #1: No

Reviewer #2: No

**Figure resubmission:**
---

## [Editor Report · Decision Letter 1]

15 Sep 2025

Dear Prof. Dias,

We are pleased to inform you that your manuscript 'InteracTor: Feature Engineering and Explainable AI for Profiling Protein Structure-Interaction-Function Relationships' has been provisionally accepted for publication in PLOS Computational Biology.

Best regards,

Fei Guo

Academic Editor

PLOS Computational Biology

Shihua Zhang

Section Editor

PLOS Computational Biology

---

## [Editor Report · Acceptance letter]

PCOMPBIOL-D-25-00691R1

InteracTor: Feature Engineering and Explainable AI for Profiling Protein Structure-Interaction-Function Relationships

Dear Dr Dias,

I am pleased to inform you that your manuscript has been formally accepted for publication in PLOS Computational Biology. Your manuscript is now with our production department and you will be notified of the publication date in due course.

With kind regards,

Judit Kozma
